# Potential Use of Vacuum Impregnation and High-Pressure Homogenization to Obtain Functional Products from Lulo Fruit (*Solanum quitoense* Lam.)

**DOI:** 10.3390/foods10040817

**Published:** 2021-04-09

**Authors:** Leidy Indira Hinestroza-Córdoba, Cristina Barrera, Lucía Seguí, Noelia Betoret

**Affiliations:** 1Grupo de Valoración y Aprovechamiento de la Biodiversidad, Universidad Tecnológica del Chocó, AA.292, Calle 22 No. 18B-10, 270002 Quibdó, Colombia; leihicor@doctor.upv.es; 2Institute of Food Engineering for Development, Universitat Politècnica de València, Camino de Vera s/n, 46022 València, Spain; mcbarpu@tal.upv.es (C.B.); lusegil@upvnet.upv.es (L.S.)

**Keywords:** LC-mass spectrometry, antioxidant capacity, vacuum impregnation, polyphenolic profile, lulo fruit juice, spermidine

## Abstract

Lulo (*Solanum quitoense* Lam.) is a Colombian fruit that is mostly used in the preparation of homemade juice as well as natural remedy for hypertension. The aim of this study was to determine physicochemical and antioxidant properties (antioxidant capacity, total phenols, flavonoids and spermidine content, and polyphenolic compounds profile by liquid chromatography—mass spectrometry (LC-MS)) of the lulo fruit and its juice. Additionally, vacuum impregnation (VI) properties of the fruit and the effect of high homogenization pressure (50, 100, and 150 MPa) on the juice properties were studied. The results revealed a good availability and impregnation capacity of the pores in fruits with similar maturity index. The main differences observed between the juice and fruit derive from removing solids and bioactive components in the filtering operation. However, the effect of high-pressure homogenization (HPH) on particle size and bioactive compounds increases the antiradical capacity of the juice and the diversity in polyphenolics when increasing the homogenization pressure.

## 1. Introduction

Having great fruit diversity, Colombia is worldwide considered one of the main producers of tropical fresh fruit with advantageous possibilities for the development of healthy food products from regional wild species [1].

One of Colombia’s territorial departments with great natural wealth is Chocó, which is characterized by its mega diverse and complex ecosystems with high primary production and little agrifood system transformation. On the other hand, according to a health situation analysis carried out and published by the Ministry of Health and Social Protection (MINSALUD) in 2018 [2], around 25.7% of the adult population in this region suffers from hypertension problems. According to the World Health Organization (WHO), noncommunicable diseases such as cancer, diabetes and hypertension are the main death causes worldwide [3]. Among them, hypertension has become the leading cause of death not only in Colombia but around the world, where it has reached up to 1.13 million sick people. In this context, the development of functional foods to prevent cardiovascular risk from highly consumed local fruits is a challenge.

Lulo (*Solanum quitoense* Lam.) is one of the most important tropical fruits in Colombia. Known as naranjilla in Ecuador and as lulum in Peru, this crop from the Solanaceae family is grown in Colombia, Ecuador, Peru, Venezuela, Guatemala, Mexico, Costa Rica, Dominican Republic, and Panama. The plant produces a spherical fruit with a diameter that varies between 3 and 8 cm [4]. In the last years, the lulo fruit has raised considerable interest in the global market due to its organoleptic characteristics, pleasant aroma, and acidic and refreshing taste [5]. Consequently, its exportation has undergone a remarkable increase, particularly reaching the United States and Europe. The fruit has considerable nutritional potential due to its high content of vitamins such as thiamin, riboflavin, vitamin A, proteins, minerals, and spermidine [6]. A recent study has demonstrated the potential antihypertensive use of lulo [7]. For the first time, these authors found *N*^1^,*N*^4^,*N*^8^-tris (Dihydrocafeoil) spermidine and *N*^1^,*N*^8^-bis-(Dihydrocafeoil) spermidine to be bitter bioactive amines in lulo fruit samples. These compounds confer some functional characteristics to lulo for the control of hypertension.

Technological transformation processes are needed to produce innovative fruit derivatives with a long shelf life and, as much as possible, preserved functional attributes and acceptable physical and sensorial characteristics. In this sense, some nonthermal technologies allow for the improvement or maintenance of relevant properties. Compared to traditional heat treatments, the application of high- and/or moderate- pressure homogenization (HPH) to fruit juices has been shown to be less destructive for food compounds of low molecular weight. Likewise, this result may be related to sensory and nutritional qualities and to the sufficient inactivation of different microorganisms. Moreover, it can improve the efficiency and performance of other processes when applied as pretreatment [8,9].

Additionally, food engineering operations like vacuum impregnation (VI) allow for the incorporating of physiologically active compounds to fruit matrices; the structure of which performs as a natural protection, thus resulting in fresh functional foods [10,11]. Fruit juices with improved properties have been incorporated into adequate fruit matrices, leading to functional foods with better composition and high potential to soften some health problems [12].

Due to the valuable and characteristic aroma of lulo fruit and its richness in antioxidant components, some studies regarding volatile profile, antioxidant capacity, and composition in carotenoids, polyphenols, or other micronutrients have been done. In addition, it is possible to find some works in the literature focused on pulp-drying to obtain a powder for industrial use. However, as far as the authors know, no work has been carried out determining the possibilities of using the fruit in VI treatments or analyzing the effect of HPH on the physicochemical and antioxidant properties of juice.

Considering everything mentioned above, it was considered relevant to study physicochemical and antioxidant properties (antioxidant capacity, total phenols, flavonoids and spermidine contents, and polyphenolic compounds profile) of the lulo fruit and its juice. Additionally, VI properties of the fruit and the effect of HPH (50, 100, and 150 MPa and one pass) on the lulo juice properties were studied too. The end goal is to provide knowledge contributing to the use of the indigenous agrifood resources of the region of Chocó (Colombia) in order to improve the health of its population.

## 2. Materials and Methods

### 2.1. Food Materials and Sample Preparation

Colombian fresh lulo fruits (*Solanum quitoense* Lam.) were purchased at the Central Market of Valencia (Spain) and stored at 4 °C until processing.

The whole fruits were processed as follows for their analysis: after removing the peduncle, they were washed and cut into pieces, which were crushed in a mortar. This whole crushed fruit is hereafter called “fruit”.

The procedure for obtaining the lulo juice was as follows: after removing the peduncle, the whole fresh fruit was washed and blended (Phillips Advance Collection Standmixer, 800 W, 2 L) for 10 min. Then, the blend was filtered and sieved with a 500 μm stainless-steel mesh sift (200/50, CISA). The filtered liquid thus obtained is hereafter called “juice”.

The sample preparation for VI experiments was as follows: the fruit was washed and cut transversally into 5 mm thickness and 64 mm diameter slices (FAGOR CF-150 slicer). The fruit skin was not removed, so as to prevent any absorption of materials through the lateral surface of the slices, which allowed assuming unidirectional matter flow. The impregnation experiments were carried out on three lulo fruit batches at a very similar ripeness stage, and three samples per batch were characterized. 

Figure 1 summarizes in a flow chart the sample preparation and treatments.

### 2.2. HPH Treatment

Juice samples were homogenized with one pass (Gea Niro Soavi-Panda Plus 2000 homogenizer, Parma, Italy) at 50, 100, and 150 MPa. Homogenized and nonhomogenized juices were refrigerated at 4 °C under aseptic conditions until later analysis.

### 2.3. VI Experiments

The VI experiments were carried at a pilot plant scale in the Institute of Food Engineering for Development at Universitat Politècnica de València (Spain). The equipment used was described by Fito et al. [13]. In all trials, the sliced fruit samples were immersed in isotonic sucrose aqueous solutions (a_w_ = 0.994 ± 0.003), and a vacuum pressure of 50 mbar was applied for 10 min. Subsequently, atmospheric pressure was restored, keeping the fruit slices immersed in the impregnation liquid for 10 min more. The impregnation parameters allow us to quantify the volumetric fraction of liquid incorporated into the porous structure and the volumetric deformation of the sample after the vacuum stage (X_1_ and γ_1_) and at the end of the process (X and γ). The effective porosity (ε_e_) provides information on the volumetric fraction of pores that are filled during the VI experiments. All the parameters were calculated using the model equations proposed by Fito et al. [14].

### 2.4. Physicochemical Characterization

In fruit, the moisture content was quantified by vacuum drying at 60 °C until constant weight [15]. Water activity was measured using a dew point hydrometer (DECAGÓN Aqualab CX-2, ±0.003, Pullman, WA, USA). Brix were determined in a refractometer (ABBE ATAGO BT, NAR T3, Tokyo, Japan) at 20 °C. pH values were measured with a potentiometer (Mettler Toledo Inlab, Schwerzenbach, Switzerland) at 20 °C. The apparent density was obtained through the volume displacement method, using a solids pycnometer and toluene as reference liquid.

In homogenized and nonhomogenized juice, density was determined with a liquid’s pycnometer. Particle size distribution was determined using a Malvern Mastersize 2000 system (Malvern Instruments Limited, Worcesterhire, UK) equipped with a blue light source (470 nm wavelength; 0.02–200 micron measuring range). A small amount of sample was diluted in deionized water in the diffractometer cell under moderate agitation until it reached 8–9% darkness. The refractive index values of the juice (cloud) and the dispersant (water) were 1.5 and 1.33, respectively. These measurements were taken using a short-wavelength blue light source in conjunction with forward and backscatter detection to enhance sizing performance in the 0.01–1000 µm range. The particle size distribution of the juice was characterized by percentages in volume (D[4,3]) and in area (D[3,2]) based diameters, and by percentiles d_10_, d_50_, and d_90_, which represent the characteristic diameters under which 10%, 50%, and 90% of the particles are within the distribution. Each analysis was repeated ten times. The rheological behavior was determined by obtaining a flow curve from a rotary rheometer (HAKKE RheoStress 1—RS1 Thermo Electron Corporation, Karlsruhe, Germany), using a Z34 DIN coaxial cylinder sensor system and a temperature bath at 20 ± 1 °C (HAKKE Phoenix 2 controller, Thermo Electron Corporation, Kalsruhe, Germany). The samples were subjected to three ascending and three descending sweeps with a velocity gradient from 0 to 300 s^−1^. Since all the samples exhibited a non-Newtonian pseudoplastic behavior, samples’ flow behaviors were modeled using the Ostwald–de Waele model.
$$\sigma =K\cdot \dot{\gamma^{n}}$$

The parameters K (consistency index (*Pa*⋅*s^n^*)) and n (flow behavior index (dimensionless)) for the model were obtained by regression using the software HAAKE RheoWin Data Manager v.3.61.0004. The results stated are the average of triplicates.

Color coordinates were obtained through a reflection spectrum between 400 and 700 nm, using a MINOLTA brand spectrocolorimeter (Model CM-3600D, Minolta, Osaka, Japan) with D65 illuminant and a 10 °C observer as references. The resulting CIE-L*a*b* color coordinates allowed for the calculating of the psychometric coordinates: tone (h*ab) and chrome (C*ab). The color difference (ΔE) between each homogenized juice and the non-homogenized one (reference) was calculated. All determinations were made by triplicate.

### 2.5. Antioxidant Properties

Antioxidant capacity by DPPH and ABTS methods, total phenols, flavonoids and spermidine content, and polyphenolic compounds profile of the lulo fruit and homogenized and nonhomogenized juices were determined.

For determination of total phenols and flavonoids content and antioxidant capacity by DPPH and ABTS method, antioxidants were extracted from fruit or from juice by diluting the samples in an 80:20 (*v/v*) methanol-water solution at a 1:10 ratio (*w/v*) and centrifuged at 10,000 rpm and 20 °C for 5 min (Selecta, “Medifriger BL-S”). Subsequent analyses were carried out on the supernatant (extract) by triplicate.

Total phenol content was determined following the Folin–Ciocalteu method [16,17]. For this procedure, 0.125 mL of extract, 0.125 mL of Folin–Ciocalteu reagent (Sigma-Aldrich, Saint Louis, MO, USA), and 0.5 mL of double-distilled water were mixed and allowed to react for 6 min. After that, 1.25 mL of 7% (*w/v*) sodium carbonate solution and 1 mL of double distilled water were added. Absorbance was measured in a spectrophotometer (Thermo Scientific, Helios Zeta U/Vis, Loughborough, UK) at 765 nm after 90 min. A standard gallic acid curve ranging from 0 to 500 mg/L was obtained. Results were expressed in milligrams of gallic acid equivalent (GAE) per gram of sample.

Flavonoid content was determined following the method described by Luximon-Ramma et al. [18]. In this case, 1.5 mL of extract and 1.5 mL of a 2% (*w/v*) aluminum chloride solution in methanol were mixed and left in the dark for 10 min. Absorbance was measured on a spectrophotometer (Thermo Scientific, Helios Zeta U/Vis, Loughborough, UK) at 368 nm. The resulting data were compared to a standard quercetin curve ranging from 0 to 350 mg/L. The results were expressed in milligrams of quercetin equivalent (EQ) per gram of sample.

Antioxidant capacity by DPPH (2.2-diphenyl-1-picrylhydrazyl) radical was determined following the method described by Kuskoski et al. [19] and Stratil et al. [20], with some modifications. A blend made up of 0.1 mL of the extract, 0.9 mL of methanol, and 2 mL of a 100 μM methanol–DPPH (39.4 μg/mL) solution was prepared. After 60 min of reaction time, absorbance was measured at 517 nm in a spectrophotometer (Thermo Scientific, Helios Zeta UV/Vis, Loughborough, UK). The results were expressed as milligrams of Trolox equivalent (TE) per gram of sample, using the Trolox calibration curve within a 0 to 500 mg/L concentration range.

Antioxidant capacity by ABTS (2.2′-azino-bis-3-ethylbenzothiazoline-6-sulfonic acid) radical was evaluated following the method described by Re et al. [21]. A solution containing 7 mM of ABTS radical and 2.45 mM of potassium persulfate was prepared and left in the dark at room temperature for 16 h. ABTS^+^ was mixed with phosphate buffer to reach an absorbance of 0.70 ± 0.02 at 734 nm. Then, 0.1 mL of extract was added to 2.9 mL of ABTS^+^ solution. Absorbance was measured at 734 nm in a spectrophotometer (Thermo Scientific, Helios Zeta UV/Vis, Loughborough, UK) after 0, 3, and 7 min of reaction time. The results were expressed as mg of Trolox equivalent (TE) per gram of sample.

Polyphenolic compounds profile and spermidine content were determined by liquid chromatography—mass spectrometry (LC-MS) analysis. First, phenolic compounds were extracted according to the procedure described by Rodrigues et al. [22] and Svobodova [23], with some modifications. In brief, 5 g of sample were mixed with 20 mL of methanol/water (80:20 *v/v*) solution by stirring (Ultra-Turrax, Staufen, Germany) at 150 rpm and room temperature for 1 h. This mix was centrifuged (Beckman Coulter AvantiTM J-25, Hamburg, Germany) at 3864× *g* and 20 °C for 5 min and the supernatant taken. The extraction procedure was repeated five times. Finally, the supernatant was filtered using a Whatman No. 1 paper filter and, subsequently, a 0.45 µm nylon filter and then directly injected into the HPLC equipment.

The equipment used for separation and identification of phenolic compounds was an (Agilent 1290 HPLC Technologies series infinity System LC, Santa Clara, CA, USA) system with a MS detector and a C18 (1.7 µm, 2.1 × 50 mm, Waters) UHPLC (Ultra High-Performance Liquid Chromatography) column. A flow rate of 0.4 mL/min and an injection volume of 5 µL at 30 °C were applied. The solvents employed were 0.1% formic acid in water (A) and 0.1% formic acid in methanol (B). The applied gradient elution was 10% B (0 min), 10% B (5 min), 100% B (12 min); 100% B (18 min), 10% B (18.5 min), 10% B (25 min). An automated calibration was performed using an external calibrant delivery system (CDS) which infuses calibration solution prior to sample introduction. The selected system was an (AB SCIEX Triple TOF™ 5600 MS, Santa Clara, CA, USA) which was used for data acquisition in both positive and negative modes over a mass range of 80–1000 m/z, under the following conditions: both negative and positive ion modes; ion source gas 1 (GC1): 55 psi; ion source gas 2 (GC2): 55 psi; gas curtain 1:25 psi at 400 °C; negative ion spray voltage (ISVF): −4500; collision energy (CE): −50; positive ion spray voltage (ISVF): 5500; collision energy (CE): 30; accumulation time for both positive and negative modes set at 100 ms. The MS used the IDA (Information-Dependent-Acquisition) acquisition method, the survey scan type (TOF-MS, Time of Fligh-Mass- Spectrometry) and the dependent scan type (product ion) with collision energy set at 50 V/30 V. Likewise, spermidine was quantified using the LC-MS/MS (Liquid Chromatography—Mass Spectrometry) method by means of direct extrapolation on the standard curve. The results were expressed in mg/L. Determinations were done by triplicate.

### 2.6. Statistical Analysis

The statistical analysis of the data was performed in a Statgraphics Centurion XVII software package, making use of simple or multifactorial analysis of variance (ANOVA) at a 95% confidence level (*p* < 0.05).

## 3. Results and Discussion

### 3.1. Vacuum Impregnation Properties of Lulo Fruit

Lulo fruit impregnation parameters provide information on the volume of external liquid that can be incorporated into the fruit tissue by controlled VI. This, in turn, informs us on the viability of incorporating protectants, preservatives, physiologically active compounds, or other additives in the porous structure of the fruit, aiming at its preservation or the formulation of new functional foods [10]. According to Fito et al. [14], the physicochemical properties of the impregnation liquid (mainly viscosity) affect the impregnation parameters, but it is the structural features of the impregnated tissue that are decisive. It is necessary that the tissue have sufficiently big intercellular and hollow spaces which, upon slicing, ensure the presence of open pores and, hence, the flow of the impregnating liquid into the porous structure of the fruit.

Table 1 shows the mean and standard deviation values of the VI parameters corresponding to three different lulo batches impregnated with an isotonic sucrose aqueous solution.

The obtained values reveal the technical feasibility of this type of unitary operation on lulo fruit. No significant differences (*p* > 0.05) were observed in parameters X_1_, γ_1_, γ and Ɛ_e_, while they were observed indeed in parameter X. A linear relation (R^2^ = 0.736) between Brix and X can be observed, allowing one to state that the riper the fruit, the higher the impregnation of its porous matrix.

The observed differences are fundamentally due to the morphological and structural variability of the fruit, which certainly deserves attention when it comes to the VI process. Positive average values for parameters X_1_ (1–5%) and X (8.6–16%) in all batches indicate the incorporation of the impregnation liquid into the porous structure of the fruit during the vacuum stage and total process, respectively. Likewise, the positive volume deformation records registered during the vacuum stage and total process, respectively, expressed by γ_1_ (3.9–7.1%) and γ (2.9–6.6%), indicate a volumetric expansion of the fruit matrix, mainly affected by vacuum application [24].

The effective porosity (Ɛ_e_) provides information on the volumetric fraction of pores that are filled during the VI experiments. It shows favorable values between 6–9%, which makes the lulo matrix appropriate for the VI process. Interesting observations result from comparing these results to the values obtained for several fruits and vegetables. The current lulo results are much lower than those reported by Betoret et al. [9] and Fito et al. [25] for Granny Smith apple (21 ± 0.9) and Soraya eggplant (64.1 ± 2), but higher than those of Chandler strawberry (6.4 ± 0.3), Hayward kiwi (0.7 ± 0.5), and Bulida apricot (2.2 ± 0.2). Regarding a likely significant batch effect (a_w_ and Brix) on Ɛ_e_, it can be observed that there are no significant differences due to the apparent homogeneity between the studied fruit samples. Differences in Brix and a_w_ may be associated with different degree of ripeness that bring with them a different structural behavior of the samples during the impregnation process. In this way, selecting fruits with the same degree of ripeness would allow obtaining fruits with a very homogeneous response to the impregnation process.

### 3.2. Physicochemical Characterization

Table 2 shows the water content, Brix, water activity, pH and density values of the studied fruit, and homogenized and nonhomogenized juice samples.

The only significant differences (*p* ≤ 0.05) are observed between fruit and juice soluble solids content (Brix) and density, regardless of HPH treatment. This is due to the fact that some fruit soluble solids remain in the bagasse after juice preparation.

Comparing the density values of homogenized and nonhomogenized juices, a slight increase is produced by the homogenization pressure. This effect can be associated to a decrease in the particle size of the suspended solids and an increase in the stability of the cloud observed in homogenized fruit juices [26,27,28].

Particle size distribution of homogenized and nonhomogenized juice samples is presented in Figure 2. A monomodal distribution ranging from 1 to 1000 µm can be observed in all juice samples, with a small irregular peak between 10 and 50 µm in the nonhomogenized juice. Although there is a remarkable reduction in particle size as a consequence of homogenization, all curves exhibit a similar particle size distribution pattern.

The above is corroborated by statistical analysis of the volume (D[4,3]) and area (D[3,2]) based diameters, which reveals a significantly negative correlation between homogenization pressure and particle size. The maximum and minimum diameters were, respectively, observed in the nonhomogenized juice (251 ± 5 µm) and the one homogenized at 150 MPa (57.94 ± 0.14 µm).

Comparing these results with those reported by Salustiana orange juice (150.1 ± 4.8, 107.7 ± 4.1 μm) and Ortanique mandarin (372.1 ± 1.9, 122.9 ± 2.2 μm) exhibits slightly similar values to those obtained for diameter (D[4,3]) in the present study. However, the (D[4,3]) and (D[3,2]) diameter values found by Castagnini et al. [29] in cranberry juice are much lower. Both authors claim particle size reduction in juices treated with homogenization pressure.

Particle size values below 10%, 50%, and 90% of the particles present in the lulo juices studied in the present work are much higher than those reported by Castagnini et al. [29] for homogenized cranberry juice.

Values of rheological properties show a pseudoplastic behavior in homogenized and nonhomogenized lulo juices (Table 2). Non-Newtonian behavior of fruit juices results from complex interactions between soluble sugars, colloidal pectic substances, and suspended solids. Pseudoplastic behavior reflects a structural reorganization of fluid particles as the velocity gradient increases, not reaching an asymptotic viscosity value. In general, in fruit juices, the higher the soluble solids content, the higher the consistency index. Nonhomogenized lulo juice have a low consistency index (K) and a lower than 1 value of the flow behavior index (n), reflecting the deviation from Newtonian behavior (n = 1) and a good capacity to be pumped and circulated in industrial plants. A significant influence of homogenization pressure on the rheology of the lulo juice as compared to the nonhomogenized one is observed. Yet, no significant differences can be observed across the homogenized samples (50 and 150 MPa). It can also be noted that there is no clear increasing or decreasing trend in the consistency index (K) after increasing the homogenization pressure.

Although sifting the juice drags a considerable amount of soluble solids, when the juice is subjected to HPH, particle size is reduced, the stability of suspended solids increased, and the juice behavior modified as if it increased in soluble solids content (higher K value). While some authors have reported similar results in terms of particle size reduction due to homogenization, others have found contrasting results for factor K. In studying the effect of homogenization on the properties of mixed peach-and-carrot juice, the authors of [26] observed a drop in the consistency index (K) and the flow behavior index. Similar trends were observed by Leite et al. [30] in orange juice and by Silva et al. [31], who studied the effect of homogenization on pineapple pulp, finding that it reduced pseudoplastic behavior (i.e., it increased the flow behavior index n and reduced the index of consistency K). Probably accounting for these contrasts, the authors of [32] have shown that the cell walls of each plant behave differently when subjected to HPH. That is to say, each fruit juice requires a different shear effort, suggesting that HPH may produce contrasting effects on different products, being mainly conditioned by the chemical nature of the components that are suspended in the juice. In studying blueberry juice, Castagnini et al. [29] obtained a consistency index of 0.57 ± 0.03 Pa.s^n^ and a flow behavior index of 0.33 ± 0.02. Chiralt et al. [33] reported a K value of 2 Pa.s^n^ and an n value of 0.43 for tomato juice with 12.8% solids and a K value of 6.48 Pa.s^n^ and an n value of 0.74 for concentrated orange juice homogenized at high pressures. According to these reported data, the consistency of homogenized and nonhomogenized lulo juice is similar to that of blueberry juice, which is of commercial use.

Table 2 shows the parameters L*, a*, b*: L* for perceptual lightness, and a* and b* for the four unique colors of human vision. Coordinate a* is relative to the green–red opponent colors, with negative values toward green and positive values toward red. The b* coordinate represents the blue–yellow opponents, with negative numbers toward blue and positive toward yellow. In addition, the psychrometric coordinates chrome (Cab*), hue (hab*) and ΔE (total color change), as functions of homogenization pressure ranging from homogenized to nonhomogenized lulo juice have been included. A slight reduction can be observed in all parameters under increased homogenization pressure.

The analysis of variance revealed a significant effect of homogenization pressure on all variables except for b* and Cab*, and L* values show significant differences between the nonhomogenized juice and the homogenized ones, the former being slightly brighter than the latter. As for parameter a*, the nonhomogenized juice exhibits the highest value. On the other hand, hue (hab*) exhibits differences between the nonhomogenized juice and the one homogenized at 50 MPa, the former one showing a less orange coloration. No significant differences were observed between the values of the studied color parameters among the homogenized juices. Thus, it can be said that juice color was not affected by pressure intensity.

Global color differences were found between the juice homogenized at 150 MPa and the nonhomogenized one used as reference. However, the visual perception of the color changes in the analyzed juices, described by the ΔE value, was not appreciated.

### 3.3. Antioxidant Properties

Figure 3 show the effect of the homogenization pressure on total phenol content, flavonoid content, and antioxidant capacity by DPPH^•^ and ABTS radical methods.

The DPPH^•^ method is more sensitive to hydrophobic flavanones, while the ABTS^+^ method is more sensitive to hydrophilic antiradicals [34]. The ABTS^+^ free radical decoloration and the DPPH^•^ free radical-scavenging assays have been reported as useful tools to evaluate antiradical activities of different fruits [9].

In comparing obtained values through statistical analysis, significant total phenol differences can be observed between the nonhomogenized juice and that homogenized at 150 MPa. Juices homogenized at 50 MPa and 100 MPa show intermediate values approaching the value of the fruit with increasing the pressure. Total flavonoids content present significant differences between fruit and juices, but this difference is more pronounced when applying the homogenization treatment.

In both cases, obtaining juice decreases the content of total phenols and flavonoids from fruit. It would be associated with the removal of the solid fraction, especially from the skin which is very rich in these compounds [35]. However, the application of an HPH produces a phenol recovery, as the pressure value applied increases while the same treatment reduces flavonoids content. As it has been explained by other authors [36], food matrix properties and processing can promote a different effect on bioactive compounds. The forces and temperature stresses created in the homogenization valve in HPH treatment could lead to a degradation of flavonoid during homogenization in this case. However, in those cases in which degradation of bioactive compounds has not been occurred yet, particle size decrease can facilitate the extraction, as it is observed in the case of phenolic compounds.

The values found in the current fruit samples do not differ much from those obtained by Contreras-Calderón et al. [1] (0.583 ± 0.024 mg GAE/g, fresh sample); by Igual et al. [37] (0.811 ± 0.016 mg GAE/g, fresh sample); and by Vasco et al. [38] (0.91 ± 0.17 mg GAE/g, fresh sample), also in lulo. On the other hand, the flavonoid content values reported by Igual et al. [37] were 0.16 ± 0.02 mg RE equivalents/g, which is a lower value than those obtained in this study (mg of quercetin equivalent).

The results of antioxidant capacity vary from 1.44 ± 0.02 mg TE/g (82.40 ± 1.41% inhibition) to 1.35 ± 0.06 mg TE/g (76.8 ± 0.8% inhibition) for the assays by DPPH^•^; and from 1.88 ± 0.03 mg TE/g (89.0 ± 1.5% inhibition) to 1.86 ± 0.01 mg TE/g (87.9 ± 0.3% inhibition) for the ABTS^+^ assays.

The values obtained by the ABTS^+^ method in both fruit and juice samples do not show significant differences. However, by the DPPH^•^ method, the juice homogenized at 100 MPa shows the highest value being significantly different from all other samples. The decrease observed in the content of total phenol and flavonoid contents in the juice compared with the fruit is not reflected when analyzing the antiradical activity. This could be explained by the fact that the liquefaction and HPH treatment may cause unbounding or chemical changes in some phenolic compounds of the fruit, giving rise to other compounds with antiradical capacity that compensate for the loss associated with sieving [39].

Authors like Forero et al. [5] and Contreras-Calderón et al. [1] in analyzing the same fruit have reported values of 71.0 ± 2.3 mg TE/g solid and 3.05 ± 0.21 mg TE/g fresh weight, respectively. The former authors obtained a fairly high value, while the latter reported a value even higher value than the one found in this study for the fruit and homogenized and nonhomogenized juices. Contrastingly, in applying the DPPH^•^ method, Vasco et al. [38] found records of 0.80 ± 0.22 mg TE/g of fresh sample, with this value being lower than those reported in this study, which vary from 1.34 to 1.44 mg TE/g of sample.

### 3.4. Profile of Phenolic Compounds by High-Performance Liquid Chromatography Coupled to Mass Spectrometry (LC-MS/MS)

Figure 4 shows the percentage distribution class of polyphenolic compounds in the lulo fruit and its homogenized and nonhomogenized juices.

A greater variety of phenolic components has been identified in homogenized juices than in nonhomogenized ones, and this variety increases with increasing homogenization pressure. Results are consistent with the results on the total phenol content previously shown in Figure 3.

In general, hydroxycinnamic acids and flavonoids were the main phenolic subclass found. It is coherent with a previous study reported by Gancel et al. [6] showing that hydroxycinnamic acids and flavonoids (i.e., quercetin glycosides, kaempferol derivatives, and 5-*O*-caffeoylquinic acid) are the dominant phenolic compounds in all parts of the lulo fruit [6].

Suárez-Jacobo et al. [40] also detected an increase in hydroxycinnamic acids in clarified apple juice when pressures of 100, 200, and 300 MPa were applied, although the general differences between the studied antioxidant activity assessment methods were not significant. Velázquez-Estrada et al. [41] have shown that orange juice homogenized at 200 and 300 MPa increased the content of flavonoids which, in this beverage with high hesperidine content, precipitate forming crystals that intertwine with proteins and other compounds.

On the other hand, it is important to mention the absence of flavonoids in the nonhomogenized lulo juice. Nevertheless, flavanones have been identified. It can be attributed to the fact that, during its elaboration, some phenolic groups (flavonoids) are bound to sugars or other molecules, thus remaining integrated in the cloud fraction and making them less accessible. Therefore, their release requires physical phenomena related to, in this case, the HPH treatment [8,42,43,44].

Some studies have shown that the Solanaceae family, especially the genus *Solanum*, are rich sources of antioxidants, such as phenolic compounds and flavonoids, and alkalis. The presence of these compounds in food has a positive impact, mainly as free radical scavengers. In addition to their nutritional benefits, they have also been associated with various biological activities such as anti-inflammatory, antitumor, and benefits related to cardiovascular diseases that include hypertension [45,46,47,48].

Appendix A, included as Appendix A present the identified compounds by LC-MS/MS in the fresh fruit extracts and their nonhomogenized and homogenized juices at different pressures and their retention time, experimental m/z, theoretical mass, molecular formula, and MS/MS fragment data. A SCIEX OS software was used for the neutral molecule and error (ppm) data, which were compared to the literature. Regarding LC-MS/MS, a total of 288 compounds were identified, including 91 hydroxycinnamic acids, 22 phenolic acids, 57 flavonoids, 38 flavanones, 21 flavones, 1 flavanol, 40 anthocyanins, 7 other phenolics, 3 stilbenes, and 4 dihydrochalcones. Qualitatively speaking, flavonoids (flavanones, flavons, and flavonols) represent the main phenolic class in this analysis, wherein 116 compounds were found in the fresh fruit and its juices homogenized at different pressures (50, 100, and 150 MPa). For their part, hydroxycinnamic acids were the second class, followed by other phenolics and phenolic acids. Hence, it can be said that fresh lulo fruit contains phenolic compounds of interest.

Appendix A shows [M-H]^−^-derived ions, MS^2^ fragments, and molecular formula for identified compounds. Compound 1 was identified as caffeic acid with [M-H]^−^ 179 and MS^2^ fragment m/z 135, 134, 89, reported by other authors [49,50,51]. Compounds 2, 3, and 4 presented the same molecular ion [M-H]^−^ 353 with a MS^2^ fragment m/z 191/179. These were described by [6,23,52,53,54,55,56].

Previous studies Park [57] showed that the presence of 5-caffeoylquinic acid and caffeic acid in fruits reduces the risk of suffering cardiovascular diseases through the suppression of *p*-selectin.

Compounds 5, 6, and 7 were assigned as *p*-coumaroylquinic acid, 4-*p*-coumaroylquinic acid, and 5-*p*-coumaroylquinic acid, respectively, with molecular mass [M-H]^-^ = 337 and with a MS^2^ fragment at m/z 191,163 as reported by Pereira et al. [58], Gutiérrez Ortiz et al. [59], Brahem et al. [52], Mikulic-Petkovsek et al. [60], Gómez-Romero et al. [55], and Rodríguez-Medina et al. [61]. Compounds 8 and 9 were, respectively, identified as quercetin-3-O-rhamnoside ([M-H]^−^ = 447 MS^2^ fragment at m/z 301) and quercetin-3-*O*-rutinoside ([M-H]^−^ = 606 MS^2^ fragment at m/z 300). These compounds have also been reported by Liu et al. [62], Kolnaik-Ostek et al. [53], Dorta et al. [63], and Fu et al. [64]. Compounds 10 and 11 correspond to isorhamnetin-3-O-rutinoside; isorhamnetin-3-*O*-rutinoside, respectively. Likewise, compounds 12 and 13 corresponding to kaempferol 3-*O*-rutinoside and kaempferol 3-*O*-glucoside; compounds 14, 15, and 16 were identified as narirutin, phloridzin, and p-coumaroyl glucose, in that order. Similarly, the compounds coumarin, sesamol, naringenin, malvidin 3-*O*-(6-acetyl-glucoside) were also identified and reported by Brahem et al. [52], Diaz-García et al. [51], Kolniak-Ostek et al. [53], and Phenol-Explorer Database [65]. Baret et al. [66] reported that polyphenols and flavonoids, such as resveratrol, quercetin, epigallocathechin-3-gallate, and curcumin, help reduce fat storage, blood pressure, blood glucose, and hemoglobin-A1c, as well as reducing insulin resistance. They showed that some compounds such as caffeic acid, gallic acid, myricetin, and catechin protect oxidative stress and help prevent cardiovascular diseases. Furthermore, Asgary et al. [67] reported that consuming pomegranate juice, which has a very similar compound to the lulo fruit juice samples studied, for 2 weeks exerts a positive effect on hypertensive individuals, and these same authors found that pomegranate juice can be considered as an effective complement to antihypertensive medications and as a component of the daily regimen for patients at high risk of hypertension.

Appendix A, also included as a Appendix A, show the polyamines identified in fruit and juice samples. The same compounds were identified by Svobodova et al. [23] in peaeggplant, Wu et al. [68] in eggplant, and Rodrigues et al. [22] in mana-cubiu.

Forero et al. [7] demonstrated the potential of the lulo as an antihypertensive due to its free bioactive amines: *N*^1^,*N*^4^,*N*^8^-tris(dihydrocaffeoyl) spermidine and *N*^1^,*N*^8^-bis(dihydrocaffeoyl) spermidine. The inhibitory activity of the fresh and dried fruit with angiotensin I-converting enzyme confirmed beneficial effect on hypertension. In another study, Gancel et al. [6] identified the same bioactive compounds in the lulo.

Polyamines are present in food, such as milk and some plants, taking part in a wide range of biologically process, such as, cellular proliferation, free radicals scavenging, the differentiation of immune cells, and neurotransmission [69].

Bomtempo et al. [70] evaluated the presence of polyamines and other bioactive amines in four varieties of the passion fruit species, reporting high spermidine concentrations and emphasizing the potential of the passifloras with functional properties relevant for the plant and human health.

However, the analytical determination of bioactive compounds in any food is required but not sufficient. Nutritional and healthy effect of food is determined by its content in macro- and micro-nutrients, their release at the target site in the adequate form, and its suitable assimilation. These three aspects considered together define the functionality of a food and are reflected separately in digestibility, bioaccessibility, and bioavailability properties. It would be necessary to carry out in vitro digestion studies or in vivo studies to quantify these properties which may be affected both by the food matrix and by the treatments applied.

## 4. Conclusions

The results obtained have shown lulo as a fruit with advantageous physicochemical and functional properties for the development of healthy food products from fresh native crops of the Colombian Pacific region. Lulo fruit can provide the bases of a new fruit juice flavor as well as other products derived from its richness in polyphenols, polyamines, and other antioxidant components.

Its structural characteristics determined indirectly from the impregnation parameters would allow the incorporation of protectants, preservatives, physiologically active compounds, or other additives. This incorporation would be slightly influenced by the maturity index but not by the variability among fruits.

From the phytochemical profile as obtained by LC-MS, 288 compounds belonging to different phenolic classes were found in the fruit and its homogenized and nonhomogenized juices (mainly flavonoids and hydroxycinnamic acids). Increasing pressure of HPH treatment increase the diversity in polyphenols from juice. Additionally, bioactive amines such as *N*^1^,*N*^4^,*N*^8^-tris(dihydrocaffeoyl) spermidine and *N*^1^,*N*^8^-bis(dihydrocaffeoyl) spermidine, whose effect against hypertension has been shown in previous studies, have been identified in fruit and juice samples.

In relation to the antiradical capacity provided by antioxidant compounds, it decreases in fresh juice compared to fruit, due to the retention of part of the solids in the filtering operation. However, it is worthwhile mentioning that the HPH at 100–150 MPa is adequate for preserving the antiradical capacity of the fruit increasing the antioxidant value of fresh juice.

However, the potential beneficial effect of the lulo fruit or of any of the analyzed juices should be assessed through in vitro studies that provide information on the bioavailability and bioavailability of the analyzed components.

## Figures and Tables

**Figure 1 foods-10-00817-f001:**
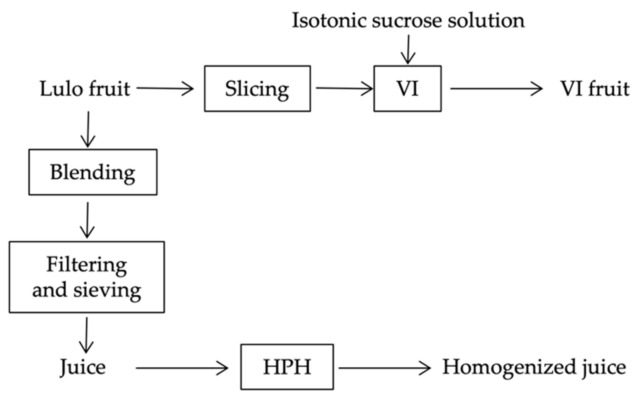
Flow chart summarizing sample preparation and treatments.

**Figure 2 foods-10-00817-f002:**
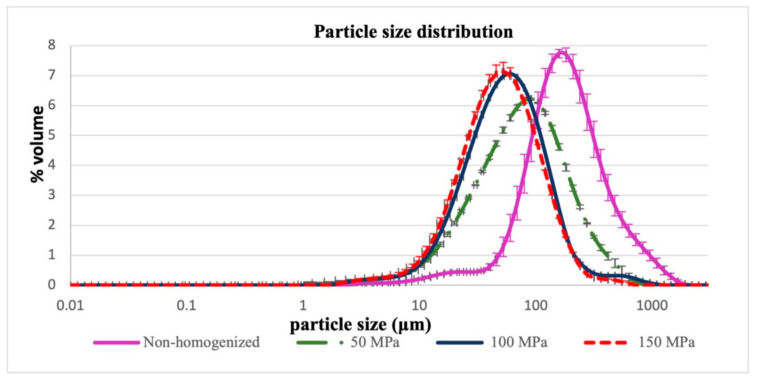
Effect of homogenization pressure on the particle size distribution of homogenized and nonhomogenized juice.

**Figure 3 foods-10-00817-f003:**
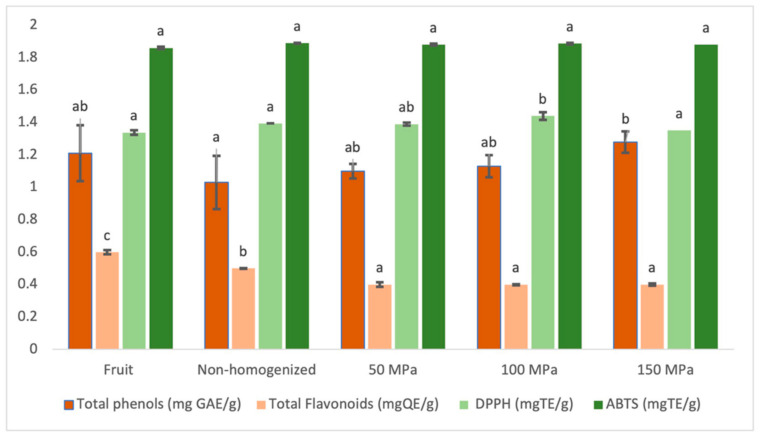
Total phenolic content, total flavonoids content, and antioxidant capacity of lulo fruit and nonhomogenized and homogenized lulo juices by the DPPH (2.2-diphenyl-1-picrylhydrazyl) and ABTS^+^ (2.2′-azino-bis-3-ethylbenzothiazoline-6-sulfonic acid) methods. Different letters for the same determination indicate significant differences (*p* ≤ 0.05).

**Figure 4 foods-10-00817-f004:**
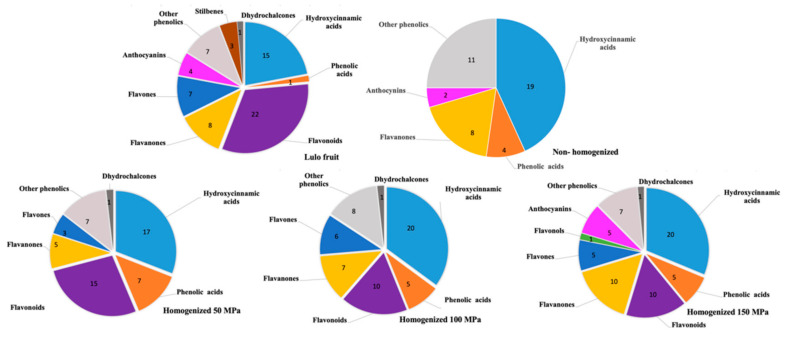
Percentage distribution of phenolic compounds identified in lulo fruit, nonhomogenized, and homogenized lulo juice.

**Table 1 foods-10-00817-t001:** Vacuum impregnation parameters of lulo fruit slices (mean ± standard deviation).

Batch	a_w_	°Brix	X_1_	γ_1_	X	γ	Ɛe
1	0.995 ± 0.001 ^a^	8.73 ± 0.06 ^a^	5 ± 7 ^a^	5 ± 4 ^a^	8.8 ± 1.6 ^a^	3 ± 3 ^a^	6 ± 4 ^a^
2	0.996 ± 0.003 ^a^	8.93 ± 0.06 ^a^	2 ± 4 ^a^	5 ± 2 ^a^	11 ± 2 ^a^	3.7 ± 0.9 ^a^	8 ± 2 ^a^
3	0.994 ± 0.002 ^a^	8.70 ± 0.17 ^a^	2.5 ± 1.3 ^a^	7.1 ± 1.0 ^a^	8.6 ± 0.9 ^a^	2.9 ± 0.8 ^a^	6.3 ± 1.2 ^a^

Different letters in the same column indicate statically significant differences (*p* ≤ 0.05).

**Table 2 foods-10-00817-t002:** Physicochemical characterization, water activity (a_w_), moisture content (x_w_), (g water/100 g), soluble solids, particle size, rheological properties, and CIE L*a*b* coordinates of the lulo fruit and homogenized and nonhomogenized juice. Mean ± standard deviation of three repetitions. (Different letters in superscripts mean significant differences (*p* < 0.05)).

	Fruit	Nonhomogenized	50 MPa	100 MPa	150 MPa
X_w_ (%)	91.2 ± 0.4	-	-	-	-
a_w_	0.994 ± 0.003 ^a^	0.994 ± 0.003 ^a^	0.997 ± 0.003 ^a^	0.995 ± 0.001 ^a^	0.996± 0.000 ^a^
Brix	8.88 ± 0.17 ^b^	6.57 ± 0.12 ^a^	6.4 ± 0.4 ^a^	6.33 ± 0.15 ^a^	6.4 ± 0.4 ^a^
pH	3.13 ± 0.16 ^a^	3.31 ± 0.01 ^a^	3.12 ± 0.02 ^a^	3.18 ± 0.03 ^a^	3.18 ± 0.03 ^a^
ρ (g/cm^3^)	1.16 ± 0.07 ^b^	1.036 ± 0.018 ^a^	1.06 ± 0.04 ^a,b^	1.07 ± 0.02 ^a,b^	1.090 ± 0.013 ^a,b^
Particle size					
D [4,3]	-	251 ± 5 ^d^	124 ± 3 ^c^	75.5 ± 1.2 ^b^	57.94 ± 0.14 ^a^
D [3,2]	-	102.3 ± 0.5 ^d^	49.9 ± 1.5 ^c^	35.28 ± 0.07 ^b^	26.83 ± 0.19 ^a^
d_10_ (μm)	-	75.8 ± 0.3 ^d^	25.7 ± 1.1 ^c^	18.94 ± 0.11 ^b^	15.63 ± 0.03 ^a^
d_50_ (μm)	-	184.2 ± 1.6 ^d^	99.82 ± 0.09 ^c^	60.01 ± 0.11 ^b^	45.02 ± 0.19 ^a^
d_90_ (μm)	-	524 ± 17 ^d^	247 ± 3 ^c^	153.7 ± 2.0 ^b^	114.2 ± 0.5 ^a^
Rheological properties					
*K* (Pa.s^n^)	-	0.39 ± 0.12 ^a^	0.9 ± 0.4 ^b^	0.79 ± 0.02 ^a,b^	1.3 ± 0.5 ^b^
*n*	-	0.44 ± 0.06 ^b^	0.37 ± 0.04 ^a^	0.37 ± 0.00 ^a^	0.34 ± 0.04 ^a^
R^2^		0.99	0.98	0.96	0.98
Color					
L*	-	40.4 ± 0.4 ^b^	40.16 ± 0.11 ^a,b^	39.409 ± 0.012 ^a^	39.44 ± 0.08 ^a^
a*	-	9.5 ± 0.2 ^b^	9.08 ± 0.07 ^a^	9.15 ± 0.01 ^a,b^	8.829 ± 0.011 ^a^
b*	-	34.9 ± 1.4 ^a^	35.4 ± 0.5 ^a^	34.54 ± 0.01 ^a^	33.8 ± 0.2 ^a^
Cab*	-	36.2 ± 1.3 ^a^	36.5 ± 0.5 ^a^	35.73 ± 0.01 ^a^	34.9 ± 0.2 ^a^
hab*	-	74.7 ± 0.2 ^a^	75.61 ± 0.09 ^b^	75.16 ± 0.01 ^a,b^	75.37 ± 0.13 ^a,b^
ΔE	-	-	0.8 ± 0.2 ^a^	1.250 ± 0.014 ^a,b^	1.74 ± 0.13 ^b^

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
