# Peer review of "Potential Use of Vacuum Impregnation and High-Pressure Homogenization to Obtain Functional Products from Lulo Fruit (*Solanum quitoense* Lam.)"

_foods, 2021, doi:10.3390/foods10040817_

Round 1
Reviewer 1 Report
Very interesting paper.
My only suggestions refers to an improved discussion of results, either in comparison with other methods, enzymes-assisted, of antioxidants by other sources and with other profiles. I list some examples, but you can find any other related:
- DOI: 10.1016/j.foodres.2017.05.017
- DOI: 10.3303/CET1438060
Regards.
Author Response
Thank you for your comments and recommendations. The authors have considered the possibility of analyzing the results from LC-MS deeper, but it would make the paper too extensive and falls outside the scope of the work.
The objective of the work is to study the physicochemical and antioxidant properties of lulo fruit and its juice, as well as its response to VI and HPH treatments with the end goal of providing knowledge contributing to the use of the indigenous agri-food resources of the region of Chocó (Colombia) in industrial applications. Affording deeper biochemistry and function of all the components identified by LC-MS as well as the study of the reasons that explain the changes induced by HPH treatment may be addressed in later more specific works.
Reviewer 2 Report
The work is aimed at defining the chemical composition and the antioxidant properties of the fruit and juice of iulo (Solanum quitoense) and the effects of homogenization pressure.
The work is sound and well described. The only shortcoming is in the fact that a chemical profile by LC-MS should afford deeper insights on the different compounds occurring in the samples. The authors describe only the class of compunds without giving information on the different constituents occurring in each class.
Author Response
Thank you for your comments and recommendations. The identification of constituents included in each class described is included in tables 3, 4, 5, 6, 7, 8 and 9 which have been provided as complementary files. Authors have considered the possibility of including all the results from LC-MS in the manuscript, but it would make the paper too extensive. Affording deeper biochemistry and function of all the components identified by LC-MS as well as the study of the reasons that explain the changes induced by HPH treatment may be addressed in later more specific works.
Reviewer 3 Report
Please consider the following comments to improve the understanding.
|
Topic 3.1/Table 1 |
The parameters X, ϒ, X1, ϒ1, and εe were not defined previously. Define these parameters in M&M section. Write how these parameters were calculated. Why the porosity was calculated here? These information must be clear. |
|
3.1 |
How did water activity and brix affect the porosity of product? Explain here. |
|
Tables |
Standardize the number of decimal places |
|
Line 279 |
To be precise the n lower than 1 represents a pseudoplastic, or shear-thinning fluid. Be clear here. Explain why |
|
281 |
In fact the pressure affected the consistency and the flow behavior indexes. Explain how important it is for practical applications. |
|
297 |
The contrasting effects with literature was attributed to the composition of raw material used. In this case, it was not HPH problem. Revise and rewrite this part. |
|
305-320 |
Insert the color ranges represented by each parameter |
|
Figure 3 |
In Lulo fruit graphic: is it ‘’2’’ or 20% in the flavonoids ? |
|
465 |
Insert the reference. |
|
|
Regarding the rheological properties, the authors must describe the power law equation and the software used for modeling in the M&M section. Provide the correlation index and mean square error of modeled data. |
Reviewer 4 Report
In this paper by Hinestroza-Cordoba, the Authors investigated the physical properties and chemical composition of lulo fruit in different form and after different technological treatments (HP and VI).
The methods used are well described and the pertinent with the goal of the study. The results have shown as the technological process may deeply affect physical and chemical properties of Lulo. Overall, this study improves the knowledge of the use of the best technological processing to ensure the highest nutritional quality of food.
Minor consideration:
I suggest including in Methods section a diagram/flow chart of the experimental design.
Line 83. Liquefied is not the most appropriate term. Please, use a better one.
Provide graphs with higher image quality and resolution. In particular, figure 3 is not comprehensible
Figure 2, legend “total phenols”. A “)” is missing
Line 328-329 and 338. As reported by the statistical symbols, there are no significant differences in total phenols between fruit and other experimental conditions. The Authors should re-write the description of these results and discussion them in a proper way.
Figure 3 is not comprehensible. Dimensions of the font are too small and not readable. I cannot review this part. Provide a higher quality image
Line 465. One or more words are missing (and also a reference)
In the study of functional food, the mere analytical determination of bioactive compounds is not sufficient. In fact, food bioactive compounds bioaccessibility is an important issue to be considered in the estimation of the overall health effect of promised functional food. Various papers have investigated as food matrix and technological processes may affect food bioactive bioaccessibility (10.1002/jsfa.10514; 10.1016/j.ifset.2020.102426). The Authors should discuss them.
Reviewer 5 Report
Betoret and his team investigated the physicochemical and antioxidant of the lulo fruit and its juice. Interestingly The main differences observed between the juice and fruit derive from removing solids and bioactive components in the filtering operation. In addition, HPH on particle size and bioactive compounds increases the antiradical capacity of the juice and the diversity in polyphenolics when increasing the homogenization pressure. The article needs following minor revision.
[1] The authors should explain (in Introduction part) how this study different from study published in following article as well as other studies on lulo fruit juices.
Gancel, A.L.; Alter, P.; Dhuique-Mayer, C.; Ruales, J.; Vaillant, F. Identifying carotenoids and phenolic 515 compounds in naranjilla (Solanum quitoense Lam. var. Puyo hybrid), an Andean fruit. J. Agric. Food Chem. 2008, 56, 11892–11899.
[1] The initial page number of ref. 6 is wrong.
[2] Line 403: “Solanum“ should be italic
[3] Line 430: “4-p-Couma-„ should be 4-p-couma-„
[4] Line 431: „5-p-Coumaroylquinic acid“ should be „5-p-coumaroylquinic acid“
[5] Lines 430-436: Authors should rephrase the following because its not clear. Better to make two or three sentences
Compounds 5, 6 and 7 were assigned as p-coumaroylquinic acid, 4-p-Couma-430 roylquinic acid, 5-p-Coumaroylquinic acid, respectively, with molecular mass [M-H]- = 431 337 with a MS2 fragment at m/z 191,163 as reported by Pereira et al. [58], Gutiérrez Ortiz 432 et al. [59] Brahem et al. [52], Mikulic-Petkovsek et al. [60], Gómez-Romero et al. [55], Ro-433 dríguez-Medina et al. [61] Compounds 8 and 9 were respectively identified as quercetin-434 3-O-rhamnoside ([M-H]-= 447 MS2 fragment at m/z 301) and quercetin-3-O-rutino-435 side ([M-H] - =606 MS2 fragment at m/z 300).
[6] Lines 465-468: This line is not clear missing the starting word(s).
[7] Page number for ref. 8 is missing.
[8] Page number for ref. 36 is missing.
